# Variational PSOM: Deep Probabilistic Clustering with Self-Organizing Maps

## Abstract

Generating visualizations and interpretations from high-dimensional data is a common problem in many fields. Two key approaches for tackling this problem are clustering and representation learning. There are very performant deep clustering models on the one hand and interpretable representation learning techniques, often relying on latent topological structures such as self-organizing maps, on the other hand. However, current methods do not yet successfully *combine* these two approaches. We present a new deep architecture for probabilistic clustering, VarPSOM, and its extension to time series data, VarTPSOM. We show that they achieve superior clustering performance compared to current deep clustering methods on static MNIST/Fashion-MNIST data as well as medical time series, while inducing an interpretable representation. Moreover, on the medical time series, VarTPSOM successfully predicts future trajectories in the original data space.

## 1 Introduction

Information visualization techniques are essential in areas where humans have to make decisions based on large amounts of complex data. Their goal is to find an interpretable representation of the data that allows the integration of humans into the data exploration process. This encourages visual discoveries of relationships in the data and provides guidance to downstream tasks. In this way, a much higher degree of confidence in the findings of the exploration is attained (Keim, 2002). An interpretable representation of the data, in which the underlying factors are easily visualized, is particularly important in domains where the reason for obtaining a certain prediction is as valuable as the prediction itself. However, finding a meaningful and interpretable representation of complex data can be challenging.

Clustering is one of the most natural ways for retrieving interpretable information from raw data. Long-established methods such as k-means (MacQueen, 1967) and Gaussian Mixture Models (Bishop, 2006) represent the cornerstone of cluster analysis. Their applicability, however, is often constrained to simple data and their performance is limited in high-dimensional, complex, real-world data sets, which do not exhibit a clustering-friendly structure.

Deep generative models have recently achieved tremendous success in representation learning. Some of the most commonly used and efficient approaches are Autoencoders (AEs), Variational Autoencoders (VAEs) and Generative Adversarial Networks (GANs) (Kingma & Welling, 2013; Goodfellow et al., 2014). The compressed latent representation generated by these models has been proven to ease the clustering process (Aljalbout et al., 2018). As a result, the combination of deep generative models for feature extraction and clustering results in a dramatic increase of the clustering performance (Xie et al., 2015). Although very successful, most of these methods do not investigate the relationship among clusters and the clustered feature points live in a high-dimensional latent space that cannot be easily visualized or interpreted by humans.

The Self-Organizing Map (SOM) (Kohonen, 1990) is a clustering method that provides such an interpretable representation. It produces a low-dimensional (typically 2-dimensional), discretized representation of the input space by inducing a flexible neighbourhood structure over the clusters. Alas, its applicability is often constrained to simple data sets similar to other classical clustering methods.

To resolve the above issues, we propose a novel deep architecture, the Variational Probabilistic SOM (VarPSOM), that jointly trains a VAE and a SOM to achieve an interpretable discrete representation while exhibiting state-of-the-art clustering performance. Instead of hard assignment of data points to clusters, our model uses a centroid-based probability distribution. It minimizes its Kullback-Leibler divergence against an auxiliary target distribution, while enforcing a SOM-friendly space. To highlight the importance of an interpretable representation for different purposes, we extended this model to deal with temporal data, yielding VarTPSOM. We discuss related work in Section 2. Extensive evidence of the superior clustering performance of both models, on MNIST/Fashion-MNIST images as well as real-world medical time series is presented in Section 4.

Our main contributions are:

- A novel architecture for deep clustering, yielding an interpretable discrete representation through the use of a probabilistic self-organizing map.
- An extension of this architecture to time series, improving clustering performance on this data type and enabling temporal predictions.
- A thorough empirical assessment of our proposed models, showing superior performance on benchmark tasks and challenging medical time series from the intensive care unit.

## 2 RELATED WORK

Self-Organizing Maps have been widely used as a means to visualize information from large amounts of data (Tirunagari et al., 2014) and as a form of clustering in which the centroids are connected by a topological neighborhood structure (Flexer, 1999). Since their early inception, several variants have been proposed to enhance their performance and scope. The adaptive subspace SOM, ASSOM (Kohonen, 1995), for example, proposed to combine PCA and SOMs to map data into a reduced feature space. (Tokunaga & Furukawa, 2009) combine SOMs with multi-layer perceptrons to obtain a modular network. (Liu et al., 2015) proposed Deep SOM (DSOM), an architecture composed of multiple layers similar to Deep Neural Networks. There exist several methods tailored to representation learning on time series, among them (Franceschi et al., 2019; Fortuin & Rätsch, 2019; Fortuin et al., 2019), which are however not based on SOMs. Extensions of SOM optimized for temporal data include the Temporal Kohonen map (Chappell & Taylor, 1993) and its improved version Recurrent SOM (McQueen et al., 2004) as well as Recursive SOM (Voegtlin, 2002). While SOM and its variants are particularly effective for data visualization (Liu et al., 2015), it was rarely attempted to combine their merits in this respect with modern state-of-the-art clustering methods, which often use deep generative models in combination with probabilistic clustering.

In particular, recent works on clustering analysis have shown that combining clustering algorithms with the latent space of AEs greatly increases the clustering performance (Aljalbout et al., 2018). (Xie et al., 2015) proposed DEC, a method that sequentially applies embedding learning using Stacked Autoencoders (SAE), and the *Clustering Assignment Hardening* method on the obtained representation. An improvement of this architecture, IDEC (Guo et al., 2017), includes the decoder network of the SAE in the learning process, so that training is affected by both the clustering loss and the reconstruction loss. Similarly, DCN (Yang et al., 2016) combines a k-means clustering loss with the reconstruction loss of SAE to obtain an end-to-end architecture that jointly trains representations and clustering. These models achieve state-of-the-art clustering performance but they do not investigate the relationship among clusters. An exception is the work by (Li et al., 2018), in which they present an unsupervised method that learns latent embeddings and discovers multi-facet clustering structure. Relationships among clusters were discovered, however, they do not provide a latent space that can be easily interpreted and which eases the process of analytical reasoning.

While there exist previous efforts to endow VAEs with a hierarchical latent space (Vikram et al., 2018; Goyal et al., 2017), to the best of our knowledge, only two models used deep generative models in combination with a SOM structure in the latent space. The SOM-VAE model (Fortuin et al., 2018), inspired by the VQ-VAE architecture (van den Oord et al., 2017) (which itself was later extended in (Razavi et al., 2019)), uses an AE to embed the input data points into a latent space and then applies a SOM-based clustering loss on top of this latent representation. It features hard assignments of points to centroids, as well as the use of a Markov model for temporal data, both of which yield inferior expressivity compared to our method. The Deep Embedded SOM, DESOM

95 (Forest et al., 2019), improved the previous model by using a Gaussian neighborhood window with
96 exponential radius decay and by learning the SOM structure in a continuous setting. Both methods
97 feature a topologically interpretable neighborhood structure and yield promising results in visual-
98 izing state spaces. However, those works did not feature empirical comparisons to state-of-the-art
99 deep clustering techniques and did not make use of many of the design principles that have recently
100 proven to be successful in this space.

101 ## 3    PROBABILISTIC CLUSTERING WITH VARIATIONAL PSOM

102 Given a set of data samples $\{x_i\}_{i=1,\dots,n}$, where $x_i \in \mathbb{R}^d$, the goal is to partition the data into a set
103 of clusters $\{S_i\}_{i=1,\dots,K}$, while retaining a topological structure over the cluster centroids.

104 The proposed architecture for static data is presented in Figure 1a. The input vector $x_i$ is embedded
105 into a latent representation $z_i$ using a VAE. This latent vector is then clustered using *PSOM*, a
106 new SOM clustering strategy that extends the *Clustering Assignment Hardening* method (Xie et al.,
107 2015). The VAE and PSOM are trained jointly to learn a latent representation with the aim to boost
108 the clustering performance. To prevent the network from outputting a trivial solution, the decoder
109 network reconstructs the input from the latent embedding, encouraging it to be as similar as possible
110 to the original input. The obtained loss function is a linear combination of the clustering loss and
111 the reconstruction loss. To deal with temporal data, we propose another model variant, which is
112 depicted in Figure 1b.

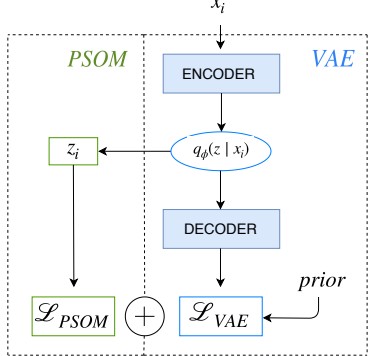

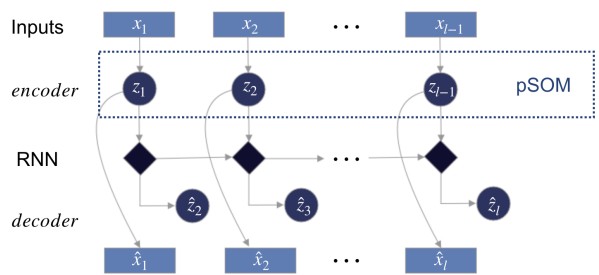

(a) VarPSOM architecture for cluster-
ing of static data. Data points $x_i$ are
mapped to a continuous embedding $z_i$
using a VAE (parameterized by $\Phi$).
The loss function is the sum of a SOM-
based clustering loss and the ELBO.

(b) VarTPSOM architecture, composed of VarPSOM modules con-
nected by LSTMs across the time axis, which predict the continu-
ous embedding $z_{t+1}$ of the next time step. This architecture allows
to unroll future trajectories in the latent space as well as the origi-
nal data space by reconstructing the $x_t$ using the VAE.

Figure 1: Model architectures of (a) VarPSOM and (b) VarTPSOM.

113 ### 3.1    BACKGROUND

114 A Self-Organizing Map is comprised of $K$ nodes connected to form a grid $M \subseteq \mathbb{N}^2$, where the
115 node $m_{i,j}$, at position $(i, j)$ of the grid, corresponds to a centroid vector, $\mu_{i,j}$ in the input space. The
116 centroids are tied by a neighborhood relation $N(\mu_{i,j}) = \{\mu_{i-1,j}, \mu_{i+1,j}, \mu_{i,j-1}, \mu_{i,j+1}\}$. Given a
117 random initialization of the centroids, the SOM algorithm randomly selects an input $x_i$ and updates
118 both its closest centroid $\mu_{i,j}$ and its neighbors $N(\mu_{i,j})$ to move them closer to $x_i$. For a complete
119 description of the SOM algorithm, we refer to the appendix (A).

The *Clustering Assignment Hardening* method has been recently introduced by the DEC model (Xie
et al., 2015) and was shown to perform well in the latent space of AEs (Aljalbout et al., 2018). Given
an embedding function $z_i = f(x_i)$, it uses a Student's t-distribution ($S$) as a kernel to measure the

similarity between an embedded data point $z_i$, and a centroid $\mu_j$:

$$s_{ij} = \frac{\left(1 + \|z_i - \mu_j\|^2 / \alpha\right)^{-\frac{\alpha+1}{2}}}{\sum_{j'} \left(1 + \|z_i - \mu_{j'}\|^2 / \alpha\right)^{-\frac{\alpha+1}{2}}} .$$

It improves the cluster purity by enforcing the distribution $S$ to approach a target distribution, $T$:

$$t_{ij} = \frac{s_{ij}^{\gamma} / \sum_{i'} s_{i'j}}{\sum_{j'} s_{ij'}^{\gamma} / \sum_{i'} s_{i'j'}}.$$

By taking the original distribution to the power of $\gamma$ and normalizing it, the target distribution puts more emphasis on data points that are assigned a high confidence. We follow (Xie et al., 2015) in choosing $\gamma$=2, which leads to larger gradient contributions of points close to cluster centers, as they show empirically. The resulting clustering loss is defined as:

$$\mathcal{L} = KL(T\|S) = \sum_i \sum_j t_{ij} \log \frac{t_{ij}}{s_{ij}}. \tag{1}$$

### 3.2 PROBABILISTIC SOM (PSOM) CLUSTERING

Our proposed clustering method, called PSOM, expands *Clustering Assignment Hardening* to include a SOM neighborhood structure over the centroids. We add an additional loss to (1) to achieve an interpretable representation. This loss term maximizes the similarity between each data point and the neighbors of the closest centroids. For each embedded data point $z_i$ and each centroid $\mu_j$ the loss is defined as the negative sum of all the neighbors of $\mu_j$, $\{e : \mu_e \in N(\mu_j(x_i))\}$, of the probability that $z_i$ is assigned to $e$, defined as $s_{ie}$. This sum is weighted by the similarity $s_{ij}$ between $z_i$ and the centroid $\mu_j$:

$$\mathcal{L}_{\text{SOM}} = -\frac{1}{N} \sum_i \sum_j s_{ij} \sum_{e:\mu_e \in N(\mu_j(x_i))} s_{ie} .$$

The complete PSOM clustering loss is then:

$$\mathcal{L}_{\text{PSOM}} = KL(T\|S) + \beta \mathcal{L}_{\text{SOM}} .$$

We note that for $\beta = 0$ it becomes equivalent to Clustering Assignment Hardening.

### 3.3 VARPSOM: VAE FOR FEATURE EXTRACTION

In our method, the nonlinear mapping between the input $x_i$ and embedding $z_i$ is realized by a VAE. Instead of directly embedding the input $x_i$ into a latent embedding $z_i$, the VAE learns a probability distribution $q_\phi(z \mid x_i)$ parametrized as a multivariate normal distribution whose mean and variance are $(\mu_\phi, \Sigma_\phi) = f_\phi(x_i)$. Similarly, it also learns the probability distribution of the reconstructed output given a sampled latent embedding, $p_\theta(x_i \mid z)$ where $(\mu_\theta, \Sigma_\theta) = f_\theta(z_i)$. Both $f_\phi$ and $f_\theta$ are neural networks, respectively called encoder and decoder. The ELBO loss is:

$$\mathcal{L}_{\text{ELBO}} = \sum_i \left[ -\mathbb{E}_z(\log p_\theta(x_i \mid z)) + D_{KL}(q_\phi(z \mid x_i) \| p(z)) \right] , \tag{2}$$

where $p(z)$ is an isotropic Gaussian prior over the latent embeddings. The second term can be interpreted as a form of regularization, which encourages the latent space to be compact. For each data point $x_i$ the latent embedding $z_i$ is sampled from $q_\phi(z \mid x_i)$. Adding the ELBO loss to the PSOM loss from the previous subsection, we get the overall loss function of VarPSOM:

$$\mathcal{L}_{\text{VarPSOM}} = \mathcal{L}_{\text{PSOM}} + \mathcal{L}_{\text{ELBO}} . \tag{3}$$

To the best of our knowledge, no previous SOM methods attempted to use a VAE to embed the inputs into a latent space. There are many advantages of a VAE over an AE for realizing our goals. Its prior on the latent space encourages structured and disentangled factors (Higgins et al., 2016), which could help clustering. The suitability of VAEs for anomaly detection (An & Cho, 2015) means that points with a higher variance in the latent space could be treated as less accurate and trustworthy. The regularization term of the VAE can be used to prevent the network from scattering the embedded points discontinuously in the latent space, which naturally facilitates the fitting of the SOM. To test if the use of CNNs can boost clustering performance on image data, we introduce another model variant called VarCPSOM, which uses convolutional filters as part of the VAE.

### 3.4 VARTPSOM: EXTENSION TO TIME SERIES DATA

To extend our proposed model to time series data, we add a temporal component to the architecture. Given a set of $N$ time series of length $T$, $\{x_{t,i}\}_{t=1,\ldots,T;i=1,\ldots,N}$, the goal is to learn interpretable trajectories on the SOM grid. To do so, the VarPSOM could be used directly but it would treat each time step $t$ of the time series independently, which is undesirable. To exploit temporal information and enforce smoothness in the trajectories, we add an additional loss to (3):

$$\mathcal{L}_{\text{smooth}} = -\frac{1}{NT}\sum_i \sum_t u_{i_t,i_{t+1}} \,, \tag{4}$$

where $u_{i_t,i_{t+1}} = g(z_{i,t}, z_{i,t+1})$ is the similarity between $z_{i,t}$ and $z_{i,t+1}$ using a Student's t-distribution and $z_{i,t}$ refers to the embedding of time series $x_i$ at time index $t$. It maximizes the similarity between latent embeddings of adjacent time steps, such that large jumps in the latent state between time points are discouraged.

One of the main goals in time series modeling is to predict future data points, or alternatively, future embeddings. This can be achieved by adding a long short-term memory network (LSTM) across the latent embeddings of the time series, as shown in Fig 1b. Each cell of the LSTM takes as input the latent embedding $z_t$ at time step $t$, and predicts a probability distribution over the next latent embedding, $p_\omega(z_{t+1} \mid z_t)$. We parametrize this distribution as a Multivariate Normal Distribution whose mean and variance are learnt by the LSTM. The prediction loss is the log-likelihood between the learned distribution and a sample of next embedding $z_{t+1}$:

$$\mathcal{L}_{\text{pred}} = -\sum_i \sum_t \log p_\omega(z_{t+1} \mid z_t) \,. \tag{5}$$

The final loss of VarTPSOM, which is trainable in a fully end-to-end fashion, is

$$\mathcal{L}_{\text{VarTPSOM}} = \mathcal{L}_{\text{VarPSOM}} + \mathcal{L}_{\text{smooth}} + \eta \mathcal{L}_{\text{pred}} \,. \tag{6}$$

## 4 EXPERIMENTS

First, we evaluate VarPSOM and VarCPSOM and compare them with state-of-the-art non-interpretable as well as SOM-based clustering methods on MNIST (Lecun et al., 1998) and Fashion-MNIST (Xiao et al., 2017) data. Here, particular focus is laid on the comparison of VarPSOM and the clustering models DEC and IDEC, to investigate the role of the VAE and the SOM loss. We then present visualizations of the obtained 2D representations, to illustrate how our method could ease visual reasoning about the data. Finally, we present extensive evidence of the performance of VarTPSOM on real-world complex time series from the eICU data set (Pollard et al., 2018), and illustrate how it allows visualization of patient health state trajectories in an easily understandable 2D domain. For details on the data sets, we refer to the appendix (B.1).

**Baselines** We used two different types of baselines. The first category contains clustering methods that do not provide any interpretable discrete latent representation. Those include k-means, the DEC model, as well as its improved version IDEC, whose clustering methods are related to ours. We also include a modified version of IDEC that we call VarIDEC, in which we substitute the AE with a VAE, to investigate the role of the VAE. For all these methods we use 64 clusters. In the second category, we include state-of-the-art clustering methods based on SOMs. Here, we used a standard SOM (minisom), AE+SOM, an architecture composed of an AE and a SOM applied on top of the latent representation (trained sequentially), SOM-VAE and DESOM. Finally, we create a modified version of our model, called AEPSOM, in which we substitute the VAE with an AE (similarly to VarIDEC). For all SOM-based methods we set the SOM grid size to $(8 \times 8)$. For different grid configurations we refer to the appendix, (B.3).

**Implementation** In implementing our models we focused on retaining a fair comparison with the baselines. Hence we decided to use a standard network structure, with fully connected layers of dimensions $d - 500 - 500 - 2000 - l$, to implement both the VAE of our models and the AE of the baselines. The latent dimension, $l$, is set to 100 for the VAE, and to 10 for the AEs. Since the prior in the VAE enforces the latent embeddings to be compact, it also requires more dimensions to learn

Table 1: Clustering performance of VarPSOM using 64 clusters arranged in a $8 \times 8$ SOM map, compared with baselines. The methods are grouped into approaches with no topological structure in the discrete latent space and interpretable methods using a SOM-based structure in the latent space, as well as an extension of our method using convolutional filters. Means and standard errors across 10 runs with different random model initializations are displayed.

| | MNIST | | fMNIST | |
| --- | --- | --- | --- | --- |
| | *pur* | *nmi* | *pur* | *nmi* |
| Kmeans | $0.845 \pm 0.001$ | $0.581 \pm 0.001$ | $0.716 \pm 0.001$ | $0.514 \pm 0.000$ |
| DEC | $0.944 \pm 0.002$ | $0.682 \pm 0.001$ | $0.758 \pm 0.002$ | $0.562 \pm 0.001$ |
| IDEC | $0.950 \pm 0.001$ | $0.681 \pm 0.001$ | - | - |
| VarIDEC (ours) | $\mathbf{0.961 \pm 0.002}$ | $\mathbf{0.698 \pm 0.001}$ | $\mathbf{0.765 \pm 0.003}$ | $\mathbf{0.569 \pm 0.002}$ |
| SOM | $0.701 \pm 0.005$ | $0.539 \pm 0.002$ | $0.667 \pm 0.003$ | $0.525 \pm 0.001$ |
| AE+SOM | $0.874 \pm 0.004$ | $0.646 \pm 0.001$ | $0.706 \pm 0.002$ | $0.543 \pm 0.001$ |
| SOM-VAE | $0.868 \pm 0.004$ | $0.595 \pm 0.004$ | $0.739 \pm 0.005$ | $0.520 \pm 0.003$ |
| DESOM | $0.939$ | $0.657$ | $0.752$ | $0.538$ |
| AEPSOM (ours) | $0.816 \pm 0.003$ | $0.555 \pm 0.001$ | $0.700 \pm 0.008$ | $0.493 \pm 0.008$ |
| VarPSOM (ours) | $\mathbf{0.964 \pm 0.001}$ | $\mathbf{0.705 \pm 0.001}$ | $\mathbf{0.764 \pm 0.003}$ | $\mathbf{0.571 \pm 0.001}$ |
| VarCPSOM (ours) | $\mathbf{0.980 \pm 0.001}$ | $\mathbf{0.726 \pm 0.001}$ | $\mathbf{0.783 \pm 0.003}$ | $\mathbf{0.574 \pm 0.001}$ |

a meaningful latent space. On the other hand, providing the AE models with a higher-dimensional latent space, needed for the VAE, resulted in a dramatic decrease of performance (see appendix B.2). VarCPSOM is composed of 4 convolutional layers of feature maps $[32, 64, 128, 256]$ and kernel size $3 \times 3$ for all layers. For all architectures, no greedy layer-wise pretraining was used to tune the VAE. Instead we simply run the VAE without the clustering loss for a few epochs for initialization. A standard SOM was then used to produce an initial configuration of the centroids/neighbourhood relation. Finally, the entire architecture is trained for $100,000$ iterations. To avoid fine-tuning hyperparameters, given the unsupervised setting, $\alpha$ is set to 10 for all experiments while the other hyperparameters are modified accordingly to maintain the same order of magnitude of the different loss components.

**Clustering Evaluation**    Table 1 shows the clustering quality results of VarPSOM and VarCPSOM on MNIST and Fashion-MNIST data, compared with the baselines. Purity and Normalized Mutual Information are used as evaluation metrics. We observe that our proposed models outperform the baselines of both categories and achieve state-of-the-art clustering performance.

**VarPSOM vs. IDEC**    VarPSOM is inspired by IDEC but it has two major differences. It uses a VAE instead of an AE and it improves interpretability in the latent space by adding a new loss that enforces a SOM structure. Since both VarIDEC and VarPSOM show superior clustering performance compared to IDEC and AEPSOM respectively (Table 1), we conclude that the VAE indeed succeeds in capturing a more meaningful latent representation compared to a standard AE. Regarding the second difference, the SOM structure was expected to slightly decrease the clustering performance, due to a trade-off between interpretability and raw clustering performance. However, we do not observe this in our results. Adding the SOM loss rather leads to an increase of the clustering performance. We suspect this is due to the regularization effect of the SOM's topological structure. Overall, VarPSOM outperforms both DEC and IDEC.

**Improvement over Training**    After obtaining the initial configuration of the SOM structure, both clustering and feature extraction using the VAE are trained jointly. To illustrate that our architecture improves clustering performance over the initial configuration, we plotted NMI and Purity against the number of training iterations in Figure 2. We observe that the performance is stable when increasing the number of epochs and no overfitting is visible.

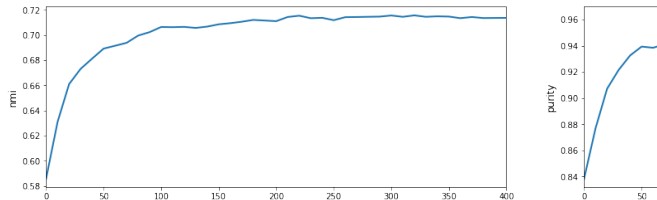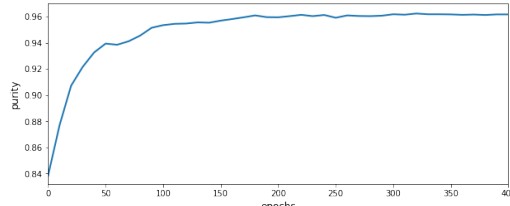

Figure 2: NMI (left) and Purity (right) performance of VarPSOM over the number of epochs on the MNIST test set.

**Role of the SOM loss**    To investigate the influence of the SOM loss component, we plot the clustering performance of VarPSOM against the weight ($\beta$) of $\mathcal{L}_{\text{SOM}}$ in Fig. 3, using the MNIST dataset. With $\beta = 30$, the $KL$ term (responsible for improving clustering purity) and the $\mathcal{L}_{\text{SOM}}$ term (responsible for enforcing a SOM structure over the centroids) are almost equal. It is interesting to observe the different trends in NMI and purity. The NMI performance increases for increasing values of $\beta$ while purity slightly decreases. Overall, enforcing a more interpretable latent space results in a more robust clustering model with higher NMI clustering performance.

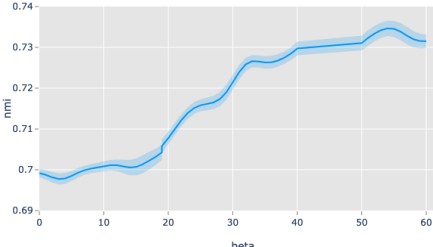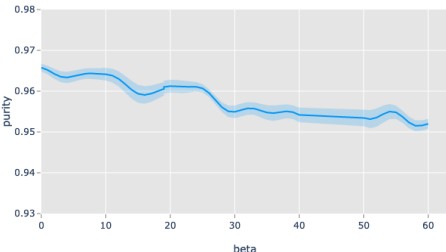

Figure 3: NMI (left) and Purity (right) performance of VarPSOM, with standard error, over $\beta$ values on MNIST test set.

**Time Series Evaluation**    We evaluate the clustering performance of our proposed models on the eICU dataset, comprised of complex medical time series. We compare them against SOM-VAE, as this is the only method among the baselines that is suited for temporal data. Table 2 shows the cluster cell enrichment in terms of NMI for three different labels, the current (APACHE-0) and worst future (APACHE-6/12 hours) physiology scores. VarTPSOM clearly achieves superior clustering performance compared to SOM-VAE. This, we hypothesize, is due to the better feature extraction using a VAE as well as the improved treatment of uncertainty using PSOM, which features soft assignments, whereas SOM-VAE contains a deterministic AE and hard assignments. Moreover, both the smoothness loss and the prediction loss seem to increase the clustering performance. More results on ICU time series are reported in the appendix (B.4).

To quantify the performance of VarTPSOM in unrolling future trajectories, we predict the final 6 latent embeddings of each time series. For each predicted embedding we reconstruct the input using the decoder of the VAE. Finally, we measure the MSE between the original input and the reconstructed inputs for the last 6 hours of the ICU admission. As baselines, we used an LSTM that takes as input the first 66 hours of the time series and then predicts the next 6 hours. Since most of the trajectories tend to stay in the same state over long periods of time, another strong baseline is obtained by duplicating the last seen embedding over the final 6 hours. The results (Table 3) indicate that the joint training of clustering and prediction used by VarTPSOM clearly outperforms the 2 baselines.

Table 2: Mean NMI and standard error of cluster enrichment vs. current/future APACHE physiology scores, using a 2D ($8 \times 8$) SOM map, across 10 runs with different random model initializations.

| Model | APACHE-12 | APACHE-6 | APACHE-0 |
|---|---|---|---|
| SOM-VAE | $0.0444 \pm 0.0006$ | $0.0474 \pm 0.0005$ | $0.0510 \pm 0.0005$ |
| VarPSOM | $0.0631 \pm 0.0008$ | $0.0639 \pm 0.0008$ | $0.0730 \pm 0.0009$ |
| VarTPSOM ($\eta = 0$) | $0.0710 \pm 0.0005$ | $0.0719 \pm 0.0006$ | $0.0818 \pm 0.0006$ |
| VarTPSOM | $\mathbf{0.0719 \pm 0.0004}$ | $\mathbf{0.0733 \pm 0.0004}$ | $\mathbf{0.0841 \pm 0.0005}$ |

Table 3: MSE for predicting the time series of the last 6 hours before ICU dispatch, given the prior time series.

| Model | LSTM | SameState | VarTPSOM |
|---|---|---|---|
| MSE | $0.0386 \pm 0.0049$ | $0.0576 \pm 0.0012$ | $\mathbf{0.0297 \pm 0.0009}$ |

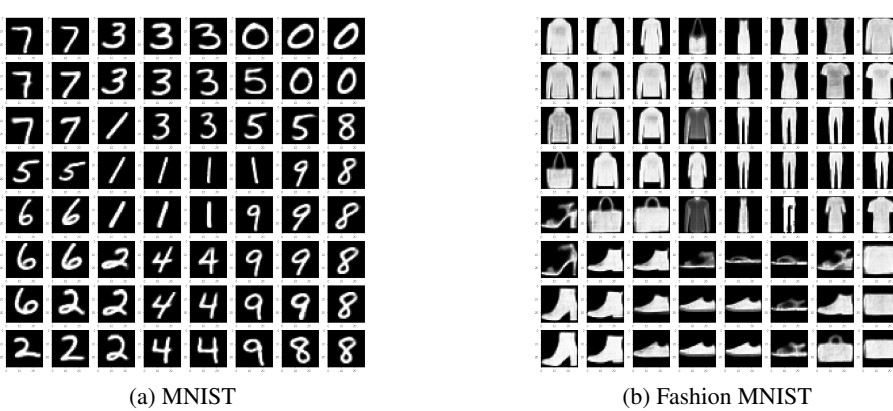

(a) MNIST         (b) Fashion MNIST

Figure 4: Reconstructions of MNIST / Fashion MNIST data from SOM cells in the 8x8 grid learned by VarPSOM, illustrating the topological neighbourhood structure induced by our method, which aids interpretability.

**Interpretability**  To illustrate the topological structure in the latent space, we present reconstructions of the VarPSOM centroids, arranged in a ($8 \times 8$) grid, on static MNIST/Fashion-MNIST data in Figure 4. On the ICU time series data, we show example trajectories for one patient dying at the end of the ICU stay, as well as two control patients which are dispatched healthily from the ICU. We observe that the trajectories are located in different parts of the SOM grid, and form a smooth and interpretable representation (Fig. 5). For further results, including a more quantitative evaluation using randomly sampled trajectories, enrichment for future mortality as well as an illustration of how the uncertainty generated by the soft assignments can help in data visualization, we refer to the appendix (B.5).

## 5   CONCLUSION

We presented two novel methods for interpretable unsupervised clustering, VarPSOM and VarTP-SOM. Both models make use of a VAE and a novel clustering method, PSOM, that extends the classical SOM algorithm to include a centroid-based probability distribution. Our models achieve superior clustering performance compared to state-of-the-art deep clustering baselines on benchmark data sets and real-world medical time series. The use of a VAE for feature extraction, instead of an AE, used in previous methods, and the use of soft assignments of data points to clusters result in an interpretable model that can quantify uncertainty in the data.

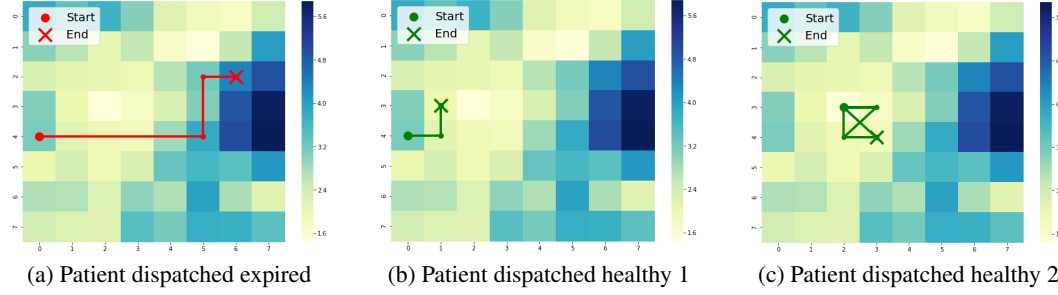

(a) Patient dispatched expired     (b) Patient dispatched healthy 1     (c) Patient dispatched healthy 2

Figure 5: Illustration of 3 example patient trajectories between the beginning of the time series and ICU dispatch, in the 2D SOM grid of VarTPSOM. The heatmap shows the enrichment of cells for the current APACHE physiology score. We observe qualitative differences in the trajectories the dying and the healthy patients.

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

# APPENDIX

## A   SELF-ORGANIZING MAPS

Among various existing interpretable unsupervised learning algorithms, Kohonen's self-organizing map (SOM) (Kohonen, 1990) is one of the most popular models. It is comprised of $K$ neurons connected to form a discrete topological structure. The data are projected onto this topographic map which locally approximates the data manifold. Usually it is a finite two-dimensional region where neurons are arranged in a regular hexagonal or rectangular grid. Here we use a grid, $M \subseteq N^2$, because of its simplicity and its visualization properties. Each neuron $m_{ij}$, at position $(i, j)$ of the grid, for $i, j = 1, \ldots, \sqrt{K}$, corresponds to a centroid vector, $\mu_{i,j}$ in the input space. The centroids are tied by a neighborhood relation, here defined as $N(\mu_{i,j}) = \{\mu_{i-1,j}, \mu_{i+1,j}, \mu_{i,j-1}, \mu_{i,j+1}\}$.

Given a random initialization of the centroids, the SOM algorithm randomly selects an input $x_i$ and updates both its closest centroid $\mu_{i,j}$ and its neighbors $N(\mu_{i,j})$ to move them closer to $x_i$. The algorithm (1) then iterates these steps until convergence.

---

**Algorithm 1** Self-Organizing Maps

---

**Require:** $0 < \alpha(t) < 1$;    $\lim_{t \to \infty} \sum \alpha(t) \to \infty$;    $\lim_{t \to \infty} \sum \alpha^2(t) < \infty$;

   **repeat**

   At each time $t$, present an input $x(t)$ and select the winner,

$$\nu(t) = \arg \min_{k \in \Omega} \|\mathbf{x}(t) - \mathbf{w}_k(t)\|$$

   Update the weights of the winner and its neighbours,

$$\Delta \mathbf{w}_k(t) = \alpha(t)\eta(\nu, k, t) [\mathbf{x}(t) - \mathbf{w}_\nu(t)]$$

   **until** the map converges

---

The range of SOM applications includes high dimensional data visualization, clustering, image and video processing, density or spectrum profile modeling, text/document mining, management systems and gene expression data analysis.

## B  EXPERIMENTAL AND IMPLEMENTATION DETAILS

### B.1  DATASETS

- **MNIST:** It consists of 70000 handwritten digits of 28-by-28 pixel size. Digits range from 0 to 9, yielding 10 patterns in total. The digits have been size-normalized and centered in a fixed-size image Lecun et al. (1998).

- **Fashion MNIST:** A dataset of Zalando's article images consisting of a training set of 60,000 examples and a test set of 10,000 examples Xiao et al. (2017). Each example is a 28×28 grayscale image, associated with a label from 10 classes.

- **eICU:** For temporal data we use vital sign/lab measurements of intensive care unit (ICU) patients resampled to a 1-hour based grid using forward filling and filling with population statistics from the training set if no measurements were available. From all ICU stays, we excluded ICU stays, which were shorter than 1 day, longer than 30 days or which had at least one gap in the continuous vital sign monitoring, which we define by a interval between 2 HR measurements of at least 1 hour. This yielded $N = 10559$ ICU stays from the eICU database. $d_{\text{vitals}} = 14$ vital sign variables and $d_{\text{lab}} = 84$ lab measurement variables were included, giving an overall data dimension of $d = 98$. The last 72 hours of these multivariate time series were used for the experiments. As labels we use a variant of the current dynamic APACHE physiology score (APACHE-0) as well as the worst APACHE score in the next 6 and 12 hours (APACHE-6/12), and the mortality in the next 24 hours. Only those variables from the APACHE score definition which are recorded in the eICU database were taken into account.

Each dataset is divided into training, validation and test sets for both our models and the baselines.

### B.2  LATENT SPACE DIMENSION

We evaluate the DEC model for different latent space dimensions. Table S1 shows that the AE, used in the DEC model, performs better when a lower dimensional latent space is used.

Table S1: Mean/Standard error of NMI and purity of DEC model on MNIST test set, across 10 runs with different random model initializations. We use 64 clusters and different latent space dimensions.

| Latent dimension | Purity | NMI |
|---|---|---|
| $l = 10$ | $0.950 \pm 0.001$ | $0.681 \pm 0.001$ |
| $l = 100$ | $0.750 \pm 0.001$ | $0.573 \pm 0.001$ |

### B.3  NUMBER OF CLUSTERS

We evaluate the NMI and purity clustering performance of our model, VarPSOM, with a varying number of clusters on the MNIST dataset. Since IDEC represents the main competitor we include it in this analysis. Figure S1 shows that VarPSOM outperforms IDEC for all the different configurations. In particular, it is interesting to observe that NMI decreases with an increasing number of clusters in both models. This is because the entropy of the clustering increases with the number of clusters.

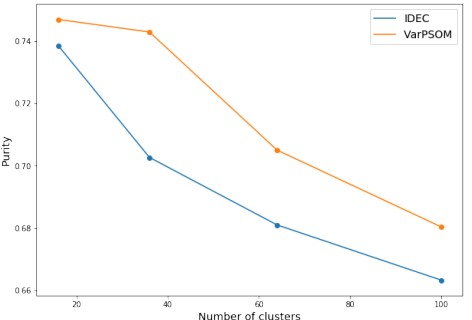 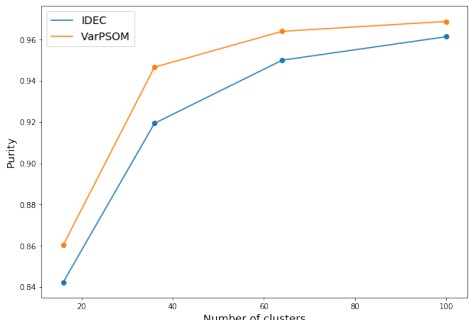

Figure S1: NMI (left) and purity (right) clustering performance of VarPSOM and IDEC with varying number of clusters on the MNIST test set.

## B.4 LEARNING HEALTH STATE REPRESENTATIONS IN THE ICU

By enforcing a SOM structure, VarPSOM, as well as VarTPSOM, project the cluster centroids onto a discrete 2D grid. Such a grid is particularly suited for visualization purposes and relations between centroids become immediately intuitive. In Fig. S2 a heat-map (colored according to enrichment in the current APACHE score, as well as mortality risk in the next 24 hours) shows compact enrichment structures. VarTPSOM succeeds in creating a meaningful and smooth neighbourhood structure. It distinguishes risk profiles with practically zero mortality risk from high mortality risk, reaching up to ≈15 %, in different regions of the map, even though it is learned in a purely unsupervised fashion. Remarkably, the two heat-maps (S2b and S2a) show different enrichment patterns. Clusters which are enriched in health states with higher APACHE scores often do not correspond exactly to clusters with a higher mortality risk. This suggests that traditional representations of physiologic values, such as the APACHE score, fail to fully use all complex multivariate relationships present in the ICU recordings, and are not associated with dynamic mortality in a simple way.

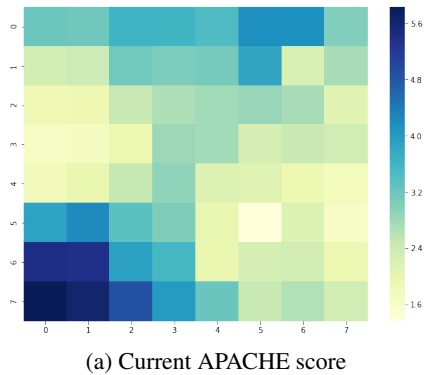

(a) Current APACHE score

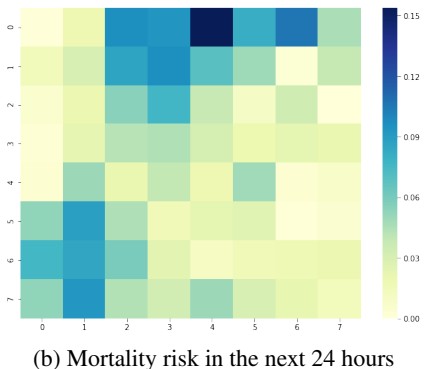

(b) Mortality risk in the next 24 hours

Figure S2: Heat-maps of enrichment in mortality risk in the next 24 hours as well as the current dynamic APACHE score, superimposed on the discrete 2D grid learned by VarTPSOM.

### B.5 VISUALIZING HEALTH STATE TRAJECTORIES IN THE ICU

To analyze the trend of the patient pathology, VarTPSOM induces trajectories on the 2D SOM grid which can be easily visualized. Fig. S3 shows 20 randomly sampled patient trajectories obtained by our model. Trajectories ending in the death of the patient are shown in red, healthily dispatched patients are shown in green.

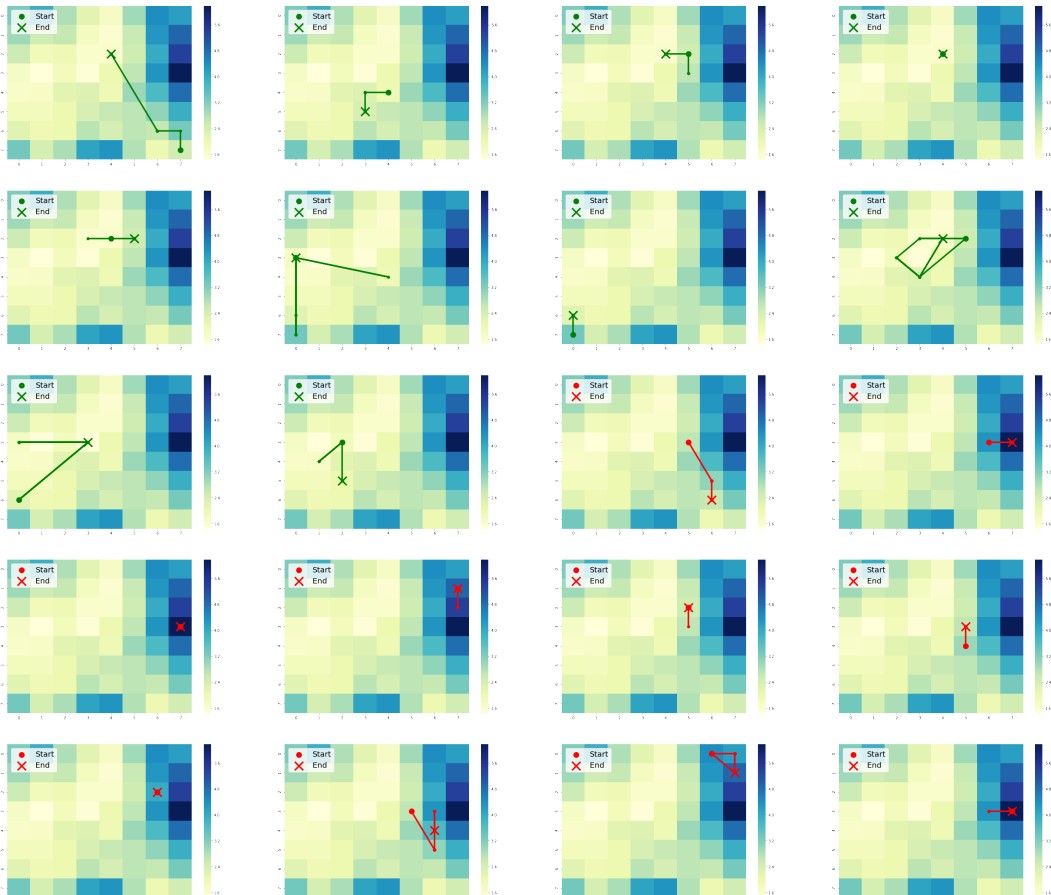

Figure S3: Randomly sampled VarTPSOM trajectories, from patients expired at the end of the ICU stay, as well as healthily dispatched patients. Superimposed is a heatmap which displays the cluster enrichment in the current APACHE score, from this model run. We observe that trajectories of dying patients are often in different locations of the map as healthy patients, in particular in those regions enriched for high APACHE scores, which corresponds with clinical intuition.

One of the main advantage of VarTPSOM over the traditional SOM algorithm is the use of soft assignments of data points to clusters which results in a better ability to quantify uncertainty in the data. For visualizing health states in the ICU, this property is very important. In Fig S4 we plot an example patient trajectory, where 6 different time-steps (in temporal order) of the trajectory were chosen. Our model yields a soft centroid-based probability distribution which evolves with time and which allows estimation of likely discrete health states at a given point in time. For each time-step the distribution of probabilities is plotted using a heat-map, whereas the overall trajectory is plotted using a black line. The circle and cross indicate ICU admission and dispatch, respectively.

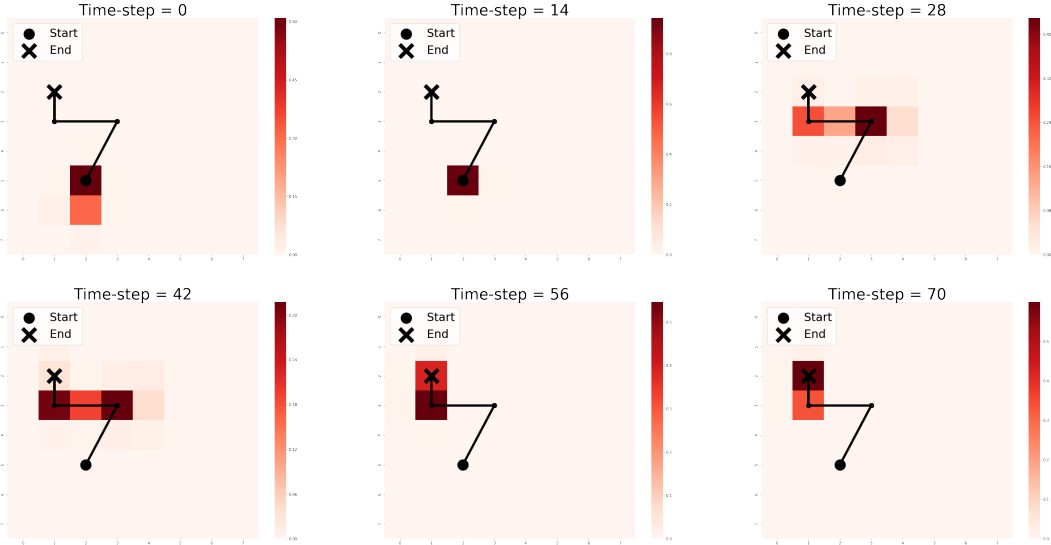

Figure S4: Probabilities over discrete patient health states for 6 different time-steps of the selected time series.

