# OpenReview forum: "Variational pSOM: Deep Probabilistic Clustering with Self-Organizing Maps"
_ICLR.cc/2020/Conference — Reject_

### Official Review · AnonReviewer1 · 2019-10-21
**Official Blind Review #1**

**Rating:** 3

**Review:**

This paper proposes VarPSOM, a method which utilizes variational autoencoders (VAEs) and clustering techniques based on self-organizing maps (SOMs) to learn clustering of image data (MNIST and Fashion MNIST in particular). An LSTM-based extension termed VarTPSOM is also evaluated on medical time series data. For the most part, the experimental results are promising, and the visualizations are particularly nice.

One of my main points of confusion is with the exposition of the method. To start, the objective presented in Eq. 3 is simply a sum of a variational lower bound and the PSOM clustering loss. Does this have a probabilistic interpretation, e.g., is it a lower bound for a particular generative model? If so, this would be useful to discuss prominently in the paper. If not, it is not clear to me what the authors are gaining from the variational framework. The paragraph at the bottom of page 4 that discusses the "advantages of a VAE over an AE" is not convincing to me. The authors claim that "points with a higher variance in the latent space could be identified as potential outliers and therefore treated as less precise and trustworthy". This isn't demonstrated in the experiments, and to the best of my knowledge, this has not been shown in prior work. If I am mistaken, a citation would be appreciated and should be included in the paper. Additionally, the claim that "the regularization term of the VAE prevents the network from scattering the embedded points discontinuously in the latent space" can also be accomplished with AEs with simple regularization, a standard technique for a wide range of AEs.

Similar comments can be made for the VarTPSOM objective in Eq. 6. Prior work in variational inference for time series, e.g., [1, 2] define a probabilistic time series generative model, from which variational inference naturally prescribes a learning objective. In my opinion, this stands in stark contrast to this work, which takes the VarPSOM objective and simply adds a time series loss on top. This is also a viable approach to building models, but why emphasize variational so much if the method is hardly motivated by anything variational?

I believe that the authors need more thorough experimental comparisons if they wish to demonstrate that their method actually benefits from the variational pieces. Most obviously, I do not believe that any of the comparisons represent the proposed method but with the VAE swapped out for some type of AE? It is my understanding that AE+SOM, SOM-VAE, and DESOM do not represent this exact ablation. VarIDEC performing better than IDEC is a data point in support of this hypothesis, however, this is a comparison of a prior method and not the proposed method.

The related work section mentions that SOM-VAE and DESOM are "likely limited by the absence of techniques used in state-of-the-art clustering methods". Is it possible to address this limitation of prior work? If so, how would this approach compare to the proposed method in terms of implementation and performance? I am not necessarily interested in an actual empirical evaluation, but including this in the related work section would likely be interesting for the reader.

The authors claim in the implementation details that "[s]ince the prior in the VAE enforces the latent embeddings to be compact, it also requires more dimensions to learn a meaningful latent space". Is there a citation for this? My understanding is that posterior collapse leads to VAEs not using additional dimensions even when they are provided, which seems to contradict this claim.

Table 2 seems to have very low NMI numbers across the board, am I reading this incorrectly? Are there prior SOTA numbers that can be included?

Finally, it seems that some of the ideas and motivation in the paper are related to learning discrete structures with variational approaches, e.g., [3, 4]. If the authors agree, it may be appropriate to include some discussion in related work.

[1] Johnson et al, "Composing graphical models with neural networks for structured representations and fast inference". NIPS 2016.
[2] Fraccaro et al, "Sequential neural models with stochastic layers". NIPS 2016.
[3] Tomczak and Welling, "VAE with a VampPrior". AISTATS 2018.
[4] Vikram et al, "The LORACs prior for VAEs: Letting the trees speak for the data". AISTATS 2019.

------

To elaborate on my "Experience Assessment" of "I have read many papers in this area": "this area" in my case refers to amortized variational inference and VAEs, not clustering techniques and SOM.

**Experience Assessment:**

I have read many papers in this area.

**Review Assessment: Checking Correctness Of Derivations And Theory:**

N/A

**Review Assessment: Checking Correctness Of Experiments:**

I assessed the sensibility of the experiments.

**Review Assessment: Thoroughness In Paper Reading:**

I read the paper at least twice and used my best judgement in assessing the paper.

---

> ### Author Response · Authors · 2019-11-12
> **Response to review #1 (Part 1/2)**
>
> Thank you for your feedback and the encouraging comments about the results / visualizations. We respond to the different points raised in the review separately below.
>
> Advantages of the Variational framework:
> We would like to clarify that our method is not only meant to cluster images, but could be used for various data types relevant in application domains, like images, time series, or others. With respect to our choice of a variational framework, there are various well known advantages of VAEs vs. deterministic AEs. Firstly, they can be more easily sampled from (which could be important when e.g synthesizing or completing time series traces forward). Further advantages include the ability to meaningfully interpolate the latent space of VAEs. Lastly, VAEs feature a more structured/disentangled latent space compared to AEs [3]. In regards to our sentence that “points with high variance could be treated as outliers”, the advantages of VAEs vs. AEs for anomaly detection has been noted in prior work (see Section 3.3 of [4], we will add this citation to the manuscript). It is true that the effect of not scattering points in latent space can also be achieved with a regularized AE, but then we would lose the advantages of VAEs as outlined above. We thank you for suggesting the additional ablation study. In response to this comment, we have added a new comparison to Table 1: We introduce the ‘AEPSOM’ model, which is identical to `VarPSOM` but substitutes the VAE with a regular auto-encoder. We observed a pronounced performance drop from 0.705 (0.571) NMI on MNIST (Fashion MNIST) to 0.555 (0.493), which suggests that a VAE-based framework has advantages when paired with the PSOM clustering loss.
>
> Time series prediction/generation:
> We would like to emphasize that the main purpose of our model is not to generate time series, hence we did not attempt to compete with complex probabilistic generative models for time series. In our experiments we want to demonstrate how the Variational PSOM, that we show to outperform a large range of baselines in the MNIST experiment, could be further extended for modeling high-dimensional real-world time series. This suggests that the model encodes information that is helpful for future prediction and thus strengthens the usefulness of the representation.
>
> Baselines vs. SOTA clustering methods:
> We apologize for the lack of clarity in the discussion of the SOM baselines and the state-of-the-art clustering methods. We meant to convey that previous deep SOM methods [1,2] did not offer comparisons to SOTA clustering methods, such as DEC and IDEC, in their respective empirical evaluations and also did not make use of modern ideas in the deep clustering field, such as the clustering assignment hardening loss. Adding those ideas to the previous approaches would lead to the different ablations of our model that we show in the experiments section (Table 1). Additionally combining this with our novel PSOM loss and the powerful latent RNN time series model then yields our proposed architecture VarTPSOM.
>
> Benefits of the VAE prior:
> The prior of the VAE leads to a smooth latent space that is shown to ease the clustering task compared to a standard Autoencoders (Table 1). One could argue that there are explicit regularizations that, if applied to a deterministic autoencoder, lead to an equally smooth and meaningful latent space [5]. However, we believe that a variational framework in which a generative model is learned from the data could represent an important step towards a more versatile and reusable framework for interpretable clustering on different types of data.
>
> SOTA for NMI on ICU time series:
> We agree that the NMI values are low compared to for instance the MNIST task. However, it should be noted that the task of assigning a physiology score to a patient is much harder, since the data is noisier and there are more different physiology scores than for instance MNIST digit classes. Also, physiology scores are naturally ordered and small differences in the score correspond to small differences in severity of physiology. To the best of our knowledge, the SOM-VAE reported SOTA results on this task [1]. We will also add K-means baseline as a reference to a revised version of our paper, which has lower performance with an NMI of 0.0411 (for APACHE 6) and 0.0384 (for APACHE 12) according to new experiment performed in response to your comment. To make the results more interpretable, we are also considering to augment them with AUROC performance on the downstream task of dynamic mortality prediction in a future version of the manuscript.

---

> > ### Author Response · Authors · 2019-11-12
> > **Response to review #1 (Part 2/2)**
> >
> > Discrete VAE priors:
> > Thank you for the pointers towards these papers. However, these related works are not concerned with clustering but only representation learning, whereas our work is targeted at providing an interpretable clustering framework through the combination of PSOM and VAE. It might be fruitful to combine some of the ideas in the suggested papers with VarPSOM in future research. We will add these references to a revised version of our manuscript.
> >
> > If you have any other suggestions on how we could improve the paper, please let us know, and we will update the manuscript if feasible.
> >
> > References
> > [1] Fortuin V, Hüser M, Locatello F, Strathmann H, Rätsch G. SOM-VAE: Interpretable Discrete Representation Learning on Time Series. arXiv preprint arXiv:1806.02199. 2018 Jun 6.
> > [2] Forest F, Lebbah M, Azzag H, Lacaille J. Deep Architectures for Joint Clustering and Visualization with Self-organizing Maps. InPacific-Asia Conference on Knowledge Discovery and Data Mining 2019 Apr 14 (pp. 105-116). Springer, Cham.
> > [3] Higgins, Irina & Matthey, Loic & Glorot, Xavier & Pal, Arka & Uria, Benigno & Blundell, Charles & Mohamed, Shakir & Lerchner, Alexander. (2016). Early Visual Concept Learning with Unsupervised Deep Learning.
> > [4] An, J., & Cho, S. (2015). Variational autoencoder based anomaly detection using reconstruction probability. Special Lecture on IE, 2(1).
> > [5] Ghosh, P., Sajjadi, M.S., Vergari, A., Black, M.J., & Schölkopf, B. (2019). From Variational to Deterministic Autoencoders. ArXiv, abs/1903.12436.
> > [6] Junyoung Chung, Kyle Kastner, Laurent Dinh, Kratarth Goel, Aaron Courville, and Yoshua Bengio. 2015. A recurrent latent variable model for sequential data. In Proceedings of the 28th International Conference on Neural Information Processing Systems - Volume 2 (NIPS'15), C. Cortes, D. D. Lee, M. Sugiyama, and R. Garnett (Eds.), Vol. 2. MIT Press, Cambridge, MA, USA, 2980-2988.
> > [7] Fraccaro et al, "Sequential neural models with stochastic layers". NIPS 2016.

---

### Official Review · AnonReviewer3 · 2019-10-23
**Official Blind Review #3**

**Rating:** 3

**Review:**

This paper aims to build a deep clustering algorithm with interpretable latent topology. The authors combine VAE, clustering assignment hardening and SOM. The authors further extend the model to deal with sequence input, by adopting temporal smoothness regularization and language model.
Good performance in clustering is achieved, experimental results also show that the topological latent structure is captured and the full (sequence) model captures more temporal information than the basic model.
However, the effect of SOM seems to be complicated and not fully explored.
Considering that VAE+SOM has been introduced before, the novelty of this paper is to combine a few techniques/tricks to strengthen the clustering performance.

Some concerns:
1.	Representation
Good clustering does not guarantee to capture a generally good representation. For example, clustering algorithms lead to a good interpretation of “color” of images that do not need to be good at clustering the semantics of images. If we consider all the factors that generate the data, it is not easy to design metrics to cluster all these factors.
2.	Vector Quantization/K-means
Can you review, compare and contrast your work with some recent discrete representation learning works? There are some works (e.g. VQ-VAE (-2)) pursing discrete representations via vector quantization or the Gumbel-softmax trick. The learned representations are also highly structured, can be used for visualization and can be used for further clustering (though might not be as good as clustering-driven approaches).
How’s the qualitative difference in the topology of centroids between your method and vector-quantization-based approaches?
3.	The role of SOM
The clustering performance mostly comes from clustering assignment hardening, while SOM does not clearly benefit the performance of clustering according to table one.
In figure three, you show that SOM’s row in affecting the clustering performance. As tuning SOM would lead to different trends in NMI and purity, what is your strategy in determining your beta?
Have you ever checked what happened to the centroid topology when varying beta?
4.	Some suggestions on experiments
.	To fully understand the benefit of each component of your loss, you can try a variant of VarTPSOM without enforcing “smoothness”
.	How is the beta affect VarTPSOM in sequence evaluation?
.	In your table one, it might be good to compare with some recent discrete latent variable models (as mentioned in bullet point one).
.       It might be good to try the number of clusters other than 64 in your experiments, towards understanding how it affects the performance.

Minor:
Denote N as the number of samples, and K as the number of centroids. At the bottom of page 3, you need O(K) to calculate each s_{i,j}. At the beginning of page 4, based on {s_{i,j}}, you either need to store intermediate computational results for acceleration, or you might need O(NK) to calculate each t_{i,j}. The whole process would be much slower than k-means/vector quantization, which seems to be not good for large enough data set.


**Experience Assessment:**

I have published one or two papers in this area.

**Review Assessment: Checking Correctness Of Derivations And Theory:**

I carefully checked the derivations and theory.

**Review Assessment: Checking Correctness Of Experiments:**

I carefully checked the experiments.

**Review Assessment: Thoroughness In Paper Reading:**

I read the paper at least twice and used my best judgement in assessing the paper.

---

> ### Author Response · Authors · 2019-11-12
> **Response to review #3**
>
> Thank you for your feedback. We respond to the different points raised in the review separately below.
>
> Novelty:
> We would like to point out that the combination VAE+SOM, trained jointly, has not been previously used, to the best of our knowledge. The SOM-VAE model, contrary to its name, uses a standard AE. Furthermore, our method allows rich data visualizations such as Appendix Figure S3 (through the use of soft assignments), where the uncertainty over different discrete states can be assessed over time. Such a visualization would not be possible using the SOM-VAE framework, which is only able to display the most likely state at any given time point.
>
> Representation vs. Clustering:
> We agree that clustering is not always the right approach for all problems, but note that it is very general, independent of tasks, and can optionally be augmented with supervised loss terms if a specific label is of special interest. However, such additions are out of scope for this work, since they are not part of the basic VarPSOM framework. Note that, for the VarTPSOM extension, by decoding the next time point in the latent space, we encourage reconstruction of the future, which can be seen as a powerful prior for a wide range of useful tasks in this domain. Overall, we believe that there are many tasks and data types where clustering can be very useful. Those are the kind of problems our method is targeted at. For instance, it has been shown that clustering can uncover medically meaningful physiological structure in ICU time series data, similar to the data we use in our experiments [1], without any supervised loss components.
>
> VQ-VAE:
> The VQ-VAE has been empirically shown to perform worse on MNIST than the SOM-VAE [2], so by transitivity it also performs worse than our proposed method. For completeness, we will add those results to a revised version of our manuscript. Also, note that in the SOM-VAE [2] paper they show that qualitatively the VQ-VAE framework makes poor use of the capacity of the latent space, in that almost all data points are assigned to a small number of clusters (see Figure 4b. in [2]).
>
> The role of SOM:
> With respect to the effect of the SOM structure on the method, since DEC/IDEC also use the clustering assignment hardening loss, this suggests that the addition of the SOM in the latent space not only improves the interpretability, but also the clustering performance itself. Figure 3 illustrates this point by investigating the effect of the SOM loss on the performance of our model.
>
> Choice of hyperparameters:
> As discussed in the manuscript, we believe that optimizing hyperparameters with respect to supervised performance measures is problematic for unsupervised learning methods, since it can overestimate the method’s performance. We thus choose our hyperparameters (including beta) in such a way that the different loss terms are of equal magnitude on average over the training set. We do not really view the number of clusters (here chosen as 64) as a hyperparameter to be optimized in our method. While it could be varied to understand its effect, we chose it a-priori because the 8x8 grid favors easy inspection by humans.
>
> Time series experiments:
> We want to point out that we have included an ablation study in the time series experiment, where we disable the smoothness loss term ($\eta = 0$). We show that this impairs the clustering performance compared to the full model (see Table 2).
>
> Minor comments:
> Thank you for these observations. In response to them we will add a discussion of the learning complexity of the algorithm to a future version of the manuscript. In this section we plan to detail potential computation bottlenecks of our method and outline solutions. Also, we aim to add pseudo-code of the training algorithm to the appendix. Generally, the average over the data set is not a computational problem, since it can either be computed exactly in a streaming fashion or approximated using a running estimate.
>
> If you have any other suggestions on how we could improve the manuscript, please let us know, and we will update the manuscript if feasible.
>
> References
> [1] Cohen MJ, Grossman AD, Morabito D, Knudson MM, Butte AJ, Manley GT. Identification of complex metabolic states in critically injured patients using bioinformatic cluster analysis. Crit Care. 2010;14:R10. doi: 10.1186/cc8864.
> [2] Fortuin V, Hüser M, Locatello F, Strathmann H, Rätsch G. SOM-VAE: Interpretable Discrete Representation Learning on Time Series. arXiv preprint arXiv:1806.02199. 2018 Jun 6.

---

> > ### Comment · AnonReviewer3 · 2019-11-14
> > **To Authors of p 618**
> >
> > Dear Authors, thanks for the very careful reply. I agree with some of your clarifications.
> >
> > See my remaining concerns please:
> >
> > Regarding your argument that SOM+VAE is actually SOM+AE, I don't quite agree.
> > Their "VAE" is not the vanilla continuous version of VAE but is a "quantized" probabilistic version.
> > Their "VAE" is more related to "VQ-VAE". Regarding the soft assignments, I think the SOM+VAE paper mentions that q(z_q|z_e) can be flexible form, but they chose "the distribution to be categorical with probability mass 1 on the closest embedding".
> > But I agree that end-2-end training with a vanilla VAE is new, and due to their choice, your SOM+VAE would be better on modeling uncertainty.
> >
> > Regarding comparing to VQ. I agree with your reasoning on why you ignored VQ-VAE.
> > I appreciate that you agree with my suggestion to including the comparison with VQ-VAE to make the table more complete.
> > Actually, I am more curious about comparing the hierarchical latent model (that is why mentioning VQ-VAE-2) with SOM. For example, an 8 by 8 hierarchical VQ/Gumbel vs your 8 by 8 SOM.
> >
> > Regarding your argument on "Choice of hyperparameters", I am not sure if I agree with it.
> > I guess I am not asking you to find the "best number of clusters".
> > We actually don't know the number of "true clusters". Varying the number of clusters is trying to see how your model works (comparing to others) in different prior assumptions.
> > I buy your argument on the beta.

---

> > > ### Author Response · Authors · 2019-11-15
> > > **Further clarifications**
> > >
> > > Thank you for your feedback. We respond to the different points raised in your response below.
> > >
> > > Comparison with SOM-VAE:
> > > We agree that the SOM-VAE is very similar to the VQ-VAE framework proposed by van den Oord et al. [2]. Nevertheless, by looking at their reconstruction loss (Equation 1 of the SOM-VAE paper [1], the squared difference between the input and the reconstruction), it is clear that they do not use a variational framework. Moreover they had to introduce an additional loss term to overcome the non-differentiability of the discrete cluster assignments (Paragraph 2.2) which is exactly the loss of a standard autoencoder. Using our probabilistic framework with soft assignments, we did not encounter any non-differentiability issues. It is true that in the methods section, they mention that the q distribution can be chosen flexibly, however there is no empirical evidence for different choices for q (like soft assignments), nor a discussion of the implications on the training process/convergence for such choices. We believe that we chose a more successful approach, incorporating a standard VAE and studying its performance. Also, we outperform their chosen (hard-clustering) form of the SOM-VAE model in 3 experiments, including the MNIST/Fashion-MNIST/eICU data sets.
> > >
> > > Comparison with VQ-VAE/VQ-VAE-2:
> > > We actually considered a comparison with the VQ-VAE-2 model. However, we noticed, as for the original VQ-VAE, the model seems to be more tailored for generation of images from latent codes rather than clustering. While the idea of coding an image at multiple scales using quantized embeddings is compelling, it is not clear how to extract one global clustering index from such multi-scale discrete latent codes. In fact, as you pointed out in your initial response, such codes would have to be post-hoc clustered and their loss function is not “clustering-driven”. We thought about some approaches that could be used, such as quantization at an arbitrary scale of the hierarchy, or joint clustering of all concatenated scales. However, we are unsure if these ad-hoc additions would be in the spirit of VQ-VAE-2, which is mainly a representation learning rather than clustering technique. We believe that it could be fruitful to introduce some of the multiscale ideas of the VQ-VAE-2 method into the PSOM in future work and to think more deeply about what a fair comparison could look like. We are not sure what you refer to by a ‘8 x 8 hierarchical VQ/Gumbel`. Do you mean 64 latent variables at a particular hierarchy of the image, or indeed 64 code-book elements? There is no topological structure among VQ-VAE-2 embeddings, so it is misleading to compare a 8 x 8 SOM grid to a 8 x 8 grid of VQ-VAE-2 latent variables, on equal terms.
> > >
> > > Effect of number of clusters:
> > > We agree with your argument that we should compare our method to others with respect to different prior assumptions (~number of clusters). Hence, we performed an additional experiment, where we compare the performance of VarPSOM with the performance of an important baseline, IDEC, across a range of different numbers of clusters. Due to lack of time, we could only perform the comparison against this baseline, but in future updates of the manuscript we will include other baselines. We will add a short section to the appendix, where we present this result and a short discussion.
> > >
> > > References
> > > [1] Fortuin V, Hüser M, Locatello F, Strathmann H, Rätsch G. SOM-VAE: Interpretable Discrete Representation Learning on Time Series. arXiv preprint arXiv:1806.02199. 2018 Jun 6.
> > > [2] van den Oord A, Vinyals O. Neural discrete representation learning. InAdvances in Neural Information Processing Systems 2017 (pp. 6306-6315).

---

### Official Review · AnonReviewer2 · 2019-10-23
**Official Blind Review #2**

**Rating:** 3

**Review:**

The paper proposes combining the latent space of a variational autoencoder with two losses that regularize the latent space. The first loss is the cluster hardening loss in Aljalbout et. al [https://arxiv.org/pdf/1801.07648.pdf]. This loss attempts to convert from a  soft-assignments of points (in latent space) to cluster centers (where the assignments are based on similarities computed via a Student t kernel) to a hard assignments of points in latent space to cluster centers. The transformation is posed as the minimization of a KL divergence.
In the model under consideration there is assumed to be a grid of "cluster centers" or centroids that all points must cluster along.

The second loss (new in this paper), penalizes the similarities between a point and a centroid from being far away from the similarities of the points to the neighbors of the centroid (on the grid). i.e. this loss tries to ensure that neighboring centroids on the grid correspond to similar points in latent space.

Finally, an extension to temporal data is proposed. The temporal model is as follows: an encoder used at each time step to obtain the latent representation and a third loss to encourage that the latent representations across consecutive time steps stays close to each other is incorporated into the learning algorithm. The latent representation is presented as input to an RNN which is used to reconstruct the data.

On NMI and cluster purity (evaluated on MNIST, fashionMNISt), the model outperforms two closely related models (the SOM-VAE and DESOM). Similarly on clustering time series data from physionet, the proposed method outperforms the SOM-VAE. The model also compares results on mean squared error (in predicting the time series 6 hours before ICU dispatch) but their baseline is an LSTM model *without* a latent variable -- a fairer baseline would be against their own model with only the variational bound used for learning. The paper also visualizes what is encoded in the centroids.

Overall this paper's contribution is the use of a VAE (rather than an autoencoder as in related work) that contains a latent space regularized to favour learning cluster structure. The paper provides good empirical evidence to suggest that the combination of the proposed losses alongside the VAE does yield better clustering performance. However, I find the addition of the two losses somewhat ad-hoc and little in the way of explanation is provided for when we should expect such a model to work and when it may not. There is not much of a discussion regarding the complexity of learning with the proposed losses but it looks like a simple algorithm for learning with Equation (1) would have to use the entire dataset to compute it. Could you comment on the scalability of the learning algorithm?

Minor comments, there are several places that need editing for grammar and context:
 * The equation at the top of page 4 needs editing within the subscripts of the summation since i is overloaded
 * line 29 talks about "the observed centroids", but centroids are not mentioned until much later in the paper
 * expand AE the first time it is used as an acronym


**Experience Assessment:**

I have published in this field for several years.

**Review Assessment: Checking Correctness Of Derivations And Theory:**

I assessed the sensibility of the derivations and theory.

**Review Assessment: Checking Correctness Of Experiments:**

I assessed the sensibility of the experiments.

**Review Assessment: Thoroughness In Paper Reading:**

I read the paper at least twice and used my best judgement in assessing the paper.

---

> ### Author Response · Authors · 2019-11-12
> **Response to review #2**
>
> Thank you for your feedback. We respond to the different points raised in the review separately below.
>
> Baseline methods:
> We would like to highlight that we do not only outperform SOM-VAE [2] and DESOM [3], but also the popular deep clustering approaches DEC and IDEC as well as the traditional clustering baseline K-means, as shown in Table 1. Different from previous works that feature a SOM-based latent space, like SOM-VAE and DESOM, we compare against state-of-the-art deep clustering methods (like those having the highest NMI in clustering MNIST digits in Table 1 of the survey [1]). We are able to provide a topological neighbourhood structure on the latent space, which favors interpretability to humans, at no loss of clustering performance. This is remarkable, since it is usually believed that there exists a strict tradeoff between interpretability and raw clustering performance. To strengthen the baseline comparison further, we added a new internal ablation to Table 1 of the manuscript, which substitutes the VAE with a standard AE.
>
> Effect of SOM structure:
> With respect to the effect of the SOM structure on the method, since DEC/IDEC also use the clustering assignment hardening loss, this suggests that the addition of the SOM in the latent space not only improves the interpretability, but also the clustering performance itself. Figure 3 illustrates this point by investigating the effect of the SOM loss on the performance of our model.
>
> Time series experiment:
> Regarding the time series experiment (VarTPSOM), we agree that one could add more diverse baselines and the comparison might not be completely convincing. However, we would like to emphasize that our main goal in this experiment is not to propose a state-of-the-art time series prediction model, but rather to demonstrate that our representation model can capture information that is relevant for future predictions, in this important data type (time series). In future versions of the manuscript we aim to strengthen this experiment by performing internal ablations and including a stronger LSTM baseline.
>
> Contributions:
> We maintain that our main contribution compared to previous approaches is not only to replace the AE with a VAE, but also to propose a new probabilistic SOM loss. To the best of our knowledge, extensions to the Clustering Assignment Hardening method to include a SOM structure over the cluster centroids have not been proposed before. We also contribute a stronger time series clustering model than was provided in previous approaches. For example, the direct competitor SOM-VAE uses a relatively inflexible Markov transition model (with a fixed order of 1), whereas we achieve superior expressivity by parameterizing the transition model with a LSTM. In addition, our VarTPSOM model encourages prediction of future states by decoding the next embedding $z(t+1)$ in time-step t. While the addition of soft assignments might sound like a simple extension, we do emphasize that in the visualization of high-dimensional systems the uncertainty over discrete states enriches the visualization considerably, enabling uncertainty maps such as in Appendix Figure S3. The SOM-VAE framework does not allow such rich visualizations, as it is a hard clustering method with no notion of uncertainty and displays only the most likely state at each time point.
>
> Exposition of methods:
> We agree that the exposition and discussion of the different loss terms lacks detail in the current manuscript. Hence, we will add more explanations and intuition regarding the losses to a revised version of the manuscript. Regarding the learning complexity of the algorithm, it is true that the target distribution in Equation (1) requires a scan over the data, however the average over the data set is not a computational problem, since it can either be computed exactly in a streaming fashion once every epoch or approximated using a running estimate. Nevertheless, we will include a new paragraph in the discussion section that details computational bottlenecks and outlines of solutions. Also, we are exploring adding pseudo-code of the training algorithm to the appendix and annotating computationally expensive steps.
>
> Minor comments:
> We apologize for these oversights and will fix them in a revised version of the paper.
>
> If you have any other suggestions on how we could improve the paper, please let us know, and we will update the manuscript if feasible.
>
> References
> [1] Aljalbout, E., Golkov, V., Siddiqui, Y., Strobel, M., & Cremers, D. (2018). Clustering with deep learning: Taxonomy and new methods. arXiv preprint arXiv:1801.07648.
> [2] Fortuin V, Hüser M, Locatello F, Strathmann H, Rätsch G. SOM-VAE: Interpretable Discrete Representation Learning on Time Series. arXiv preprint arXiv:1806.02199. 2018 Jun 6.
> [3] Forest, Florent & Lebbah, Mustapha & Azzag, Hanene & Lacaille, Jérôme. (2019). Deep Embedded SOM: Joint Representation Learning and Self-Organization.

---

### Decision · Program_Chairs · 2019-12-19

**Decision:**

Reject

**Comment:**

The authors present a deep model for probabilistic clustering and extend it to handle time series data.   The proposed method beats existing deep models on two datasets and  the representations learned in the process are also interpretable.

Unfortunately, despite detailed responses by the authors, the reviewers felt that some of their main concerns were not addressed. For example, the authors and the reviewers are still not converging on whether SOM-VAE uses a VAE or an autoencoder. Further, the discussion about the advantages of VAE over AE is still not very convincing. Currently the work is positioned as a variational clustering method but the reviewers feel that it is a clustering method which uses a VAE (yes, I understand that this difference is subtle but needs to be clarified).

The reviewers read the responses of the author and during discussions with the AC suggested that there were still not convinced about some of their initial questions. Given this, at this point I would prefer going by the consensus of the reviewers and recommend that this paper cannot be accepted.